# Exploring the Critical Success Factors of Value Management Implementation for Sustainable Residential Building Project: A Stationary Analysis Approach

Ahmed Farouk Kineber [1,*] , Md Sharif Uddin [2,3] and Alaa Fouad Momena [2]

1 Department of Civil Engineering, College of Engineering in Al-Kharj, Prince Sattam Bin Abdulaziz University, Al-Kharj 11942, Saudi Arabia
2 Department of Industrial Engineering, Prince Sattam Bin Abdulaziz University, Al Kharj 11942, Saudi Arabia
3 Department of Mathematics, Jahangirnagar University, Savar 1342, Bangladesh
* Correspondence: A.farouk.kineber@gmail.com or a.kineber@psau.edu.sa

**Abstract:** During the past two decades, value management (VM), has developed into a recognized construction practice. However, the methods and activities associated with VM adopt informal approaches in developing countries. This study aims to explore the critical success factors (CSFs) of VM implementation. Consequently, VM CSFs were investigated from the previous literature and further categorized over a semi-structured interview. The importance of these CSFs investigated by 335 structured questionnaires completed by residential building professionals. Subsequently, the exploratory study using the exploratory Pearson correlation of the VM CSFs was employed to validate the categorization resulting from a semi-structured interview and pilot study phases. Based on the validation results, the VM CSFs may be divided into four dimensions: culture and environment, workshop dynamics, stakeholder and knowledge, and standardization. Through important relative index (RII) analysis, the essential CSFs creates a VM team from a variety of disciplines, VM knowledge, experience of participants, and professional experience of the different participants' diverse disciplines. In addition, this research used a stationary analytic strategy to evaluate the degree to which VM critical success factors (CSFs) have been incorporated into residential construction projects in Egypt. The results revealed that "establishing the roles and purposes of various professions" was the stationary success factor for adopting VM. This research establishes a road map for successful VM implementation via VM CSFs in Egypt and other underdeveloped nations. Stakeholders in the residential construction sector would benefit from this study by learning more about VM CSFs and how they may be used to increase the value of their projects.

**Keywords:** construction project management; building performance; value management; value engineering; residential building projects; Egypt; developing countries; critical success factors

## 1. Introduction

The residential building market is one of the most dynamic sectors in several nations. The success of residential construction projects takes a back seat to other factors in developing countries. Even though these countries have experienced rapid economic expansion, it is undeniable that the residential construction business is essential to guarantee minimum living standards for the citizens [1]. The residential construction industry has experienced radical change in many developing nations to support their economic goals [2]. It has been established that financial plans in underdeveloped nations are mostly silent during the upgrading phase [3]. Consequently, Ofori [4] noted that poor project management has led to many difficulties for construction companies operating in underdeveloped countries. The project timetable is often not met, and the final budget is much more than projected [5]. In the same vein, Kim et al. [6] acknowledged that residential development projects were suspended or abandoned because of a lack of finance. Furthermore, there has not been enough

effort to maximise investment in urban housing sectors through residential development projects in developing countries [7].

Egypt is a developing country with extreme-risk markets because of low incomes, high rates of unemployment, and security concerns [8]. Dire currency instability, a dearth of informed commercial opportunities, and the limitations of financial models all contributed to the risk [9]. Despite tremendous progress and rapid population growth between 1950 and 2020, it remained one of the largest inhabited countries in North Africa [10]. Estimates showed that by 2020, the population will be more than five times what it was in 1950. Subsequently, Egyptian policymakers are facing many challenges in achieving their residential building project requirements [11]. This challenge puts difficulty on the Egyptian government to produce sufficient residential building projects, in response to demands by increasing population [12].

Furthermore, despite the population increasing by 9% between 2008 and 2013, the increase in rural development between 2001 and 2012 boosted only by 0.9% [11]. It has highlighted the need to improve "residential building's success" by adding value, cutting costs, and increasing quality to meet the needs of Egyptian homeowners. Value management (VM) can dramatically integrate the successful approach for residential building projects [13], and VM is recommended as a mechanism for improving the success value of a project [14].

Value management is a multidisciplinary, team-oriented, organisational, and universal analytic tool exclusively designed to help clients achieve their objectives [15,16]. It recommends a technique that starts with the planning phase, ends the project, stimulates, and reduces unnecessary costs [17]. Requests are handled in the construction industry to increase productivity and measures necessary to decrease the residential building project expense. Implementation of VM is supposed to be valuable for project clients, consultants, and contractors [18]. VM has been shown to reduce the cost of investment in construction projects by 10 to 25%, as reported by Ellis et al. [19]. However, VM methods do not receive the needed attention in most developing countries, including Egypt [6]. Oke and Ogunsemi [20] establish that insignificant VM studies are continuing, and well-organised workshops are in progress in these countries. Even though many previous studies have covered the advantages, activities, and technological efficacy of VM in many other nations, no attempt has been made to quantify the extent to which VM is used in Egyptian building projects. Keeping in mind that there is a lack of study in this area is essential. This is also true of the Egyptian building sector. Abdelghany et al. [21] argued that there were no comprehensive studies conducted to determine the current state of VM education and adoption in Egypt. In Egypt, it is impossible to use the conventional VM approach. Consequently, Othman et al. [22] reported that the vast majority (86.4%) of construction industry experts do not include VM in their projects. Of course, this motivates impromptu approaches, such as disorganised teamwork, which do not help in keeping construction expenses down.

The implementation of VM is necessary to control the performance and success of residential building projects., Likewise, the critical success factors (CSFs) are crucial for implementing VM [23]. Research and analysis of this subject were invented by Romani [24]. However, Shen and Liu [17] studied CSFs by choosing various applications in the United Kingdom, Hong Kong, and the United States. Nevertheless, no data have been identified and collected on this issue from the Egyptian building industry's perspective. It is argued by Pasquire and Mauro [25] and Hunter and Kelly [26] that modifications in the political, economic, cultural, and project-delivery systems could lead to different CSFs in various geographical regions for the same industry [27]. Based on these revelations, this study hypothesised that there is a consensus on the significance of VM CSFs in residential construction projects. Hence this study was motivated by finding answers to the following research question. What are the critical VM CSFs? Consequently, this research tried to answer this question within the context of Egypt through analysing the VM workshops CSFs.

## 2. VM and the Sustainable Construction Industry

Numerous types of research have emphasised sustainability topics [28]. It is challenging to transform strategic sustainability aims and strategy procedures for projects [29]. There should be harmony between sustainability's economic, ecological, and social components [28,30]. Since sustainability has gained traction in the construction sector, businesses have looked for viable methods to incorporate it into their current infrastructure [13]. Drivers that might boost VM's massive adoption at the vital strategic phases include the necessity for sustainable improvement and the creative corporate social responsibility ethic applied via enterprises [31]. VM is well-established as a structured and analytical approach intended to enhance value for money by enabling the necessary services at the least price steady with the needed quality and sustainability [32]. Modern viewpoints suggested that VM has played a more crucial role in the early procurement stage, when it can identify, explain, and validate client expectations and goals [33]. According to this point of view, virtual reality (VR) is most aligned with the project briefing phase [34], although it is unclear how they are interpreted in practice by the vocations working in the construction industry.

In VM, study typically range from 1.5 to 5 days in length [35]. Variations in the VM scope and the VM phases number can serve as examples of the elements that affect the number of days spent in the VM workshop (s). As exemplified by the specialist of the US value engineer's society, the latter factor emphasised a three-stage methodology: pre-stage value stage, and post-stage. The second step, (i.e., value stage), proposes to adopt six phases of the study [14]. The following illustration for these phases for the workshop allows chosen team participants to refine the project in order to generate a cost-effective model for their projects [19]:

*Information Phase*—This phase sought to assist clients and end users by clearly and precisely expressing the objectives, parameters, expectations, and needs of their upcoming creativities. The current conditions of the project and the goals of the study are defined closely and reviewed in this phase [36]. Details and information on the context, design, estimated costs, role and projects budget, should be provided. Likewise, the limitations of the project are provided in this phase [37–39]. Subsequently, construction stakeholders such as the team leader can provide critical information concerning their fields [40].

*Function Phase*—This is a consistent phase that describes and fulfils the principles, needs, goals and objectives of the project, and its purpose is to develop, recognise and classify secondary and primary functions [40]. It allows team members to define and express the procedure by highlighting the function of the project [41]. The fundamental and secondary functions are then identified and classified along with the expenditures related to them.

*Creativity Phase*—The ideas are generated and developed during this phase with the aim to perform the necessary and preferred functions of the project. At this phase, innovative methods and procedures, such as synectic, side-thinking, and brainstorming are applied. As a result of preventing criticism and repression among VM team members, the VM facilitator creates a good environment. The members of the VM Workshop team can then do research, investigate, develop, and test alternative approaches and techniques to complete necessary tasks [42].

*Evaluation Phase*—This stage aims to evaluate and scrutinize the ideas and suggestions that were identified during the creativity stage. Previous, studies have found that it is best to do additional testing of recommendations and suggestions to determine how to effectively attain the project studies' targeted goals and objectives [40].

*Development Phase*—This phase addresses all of the disadvantages, benefits and potentials of the proposed concepts and ideas. It can allow VM members to generate proposals and suggestions through generating sketches, descriptions/materials, drawings, requirements and information, the team preparations, manufacturers, and producers.

*Presentation Phase*—In this phase, a proposal in the development phase shall be included in the strategic plan. Therefore, VM team members should be informed and be warned of the reluctance to ignore the step-by-step strategy of the VM study [43].

Conversely, sustainability magnitudes are routinely included in VM research, even if the term "sustainability" is not clearly mentioned [44]. Different VM studies may come up with different ideas about sustainability due to factors including the owners' goals, the importance of building and execution, the knowledge of the VM team members, and the urgency of the situation [44]. For instance, many UK developments have taken sustainability into account, including one in Crianlarich, Startfilan, where virtual machine management was used to benefit the local community [45], Stewart shire's green residential construction and services, as well as Scotland's Loch Katrine Water Project [46]. In a paper titled "Encouraging Major Clients to Implement VM Concepts", Hayles [47] argued that doing so would help establish a sustainable mechanism for making decisions within an organisation. It is clear from Al-Yousefi's [48] study that using VM as a framework to promote and launch sustainability concepts has several benefits. According to Kelly et al. [49], combining VM with sustainability is strongly encouraged by the dedication of multidisciplinary stakeholder members, coordinated and formal VM study, acceptance of sustainable principles as project objectives, and delivery of the whole project cost. Thus, implementing sustainability using VM is practical and recommended [50]. Significant progress has been made in implementing virtual reality (VR) in the building sector over the past three decades. However, there is a paucity of appropriate research about Egypt [51], and there is no study which has focused solely on comparing the present practice and application of VM by the stakeholders in the built environment. Creating an opinion poll of stakeholders in Egypt's built environment was a straightforward way to address this deficiency and close this knowledge gap. Sustainable construction was the end aim. Thus, this research was conducted by focusing on how VM CSFs are understood and implemented in the real world.

## 3. Methodology

The aim of this study is to recognise the critical VM CSFs in Egyptian construction. Thus, the research methodology summarised the procedures adopted to achieve this aim [52]. This was accomplished through reading relevant material and conducting a semi-structured interview. A pilot study through exploratory factor analysis (EFA) was performed to check the results obtained from the interview [53]. Therefore, the questionnaire survey was conducted to ask professionals with sufficient VM studies experience to express their opinion on each of the nominated factors. The Pearson correlation analysis was employed to calculate the correlation among the factors and validate the results from EFA. Moreover, the relative importance index (RII) analysis was conducted to examine the different factors and groups that are vital to implementing VM effectively in the Egyptian building industry. The research design, which is adapted from [54,55] is illustrated in Figure 1.

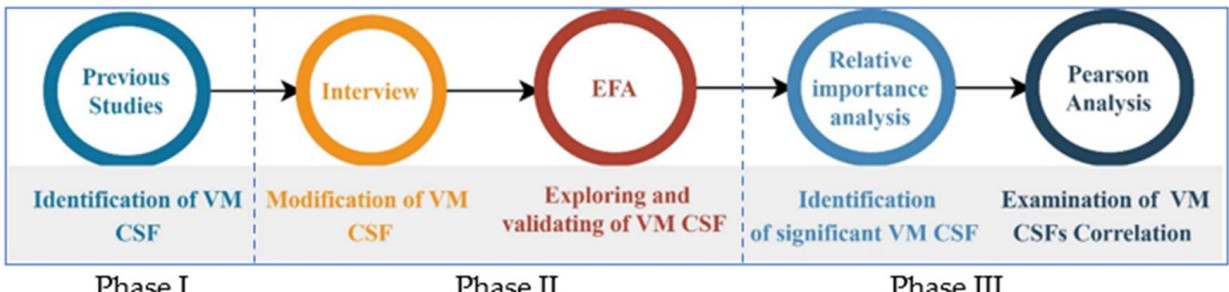

**Figure 1.** Research design.

### 3.1. Semi-Structured Interviews

Based on the suggestions made by Sanders [56] and Hesse-Biber [57], the research involved ten interviews. Hence fifteen experts on three levels were selected: (i) the years of experience, (ii) the educational attainment, and (iii) position through a "purposive

sampling" approach. Four professionals, five practitioners from the private sector and six professionals from the consultative industry, were interviewed. The interviewees have vast experience in the residential building field ranging from nine and 40 years, and the contributors were selected based on the following criteria: experience, education, and work. Likewise, four academics, five private industry practitioners, and six consultants were interviewed, who held several positions, including site engineer, consultant, project manager, executive director, and the manager following Othman et al. [58]. Their primary roles cover all key stakeholders, clients, suppliers, or contractors in the building industry, ensuring extensive experience from various perspectives. Their experience also included working with government, private sector, and self-employed agencies. Consequently, the interviewed experts appealed that a more proper system must champion the VM implementation in projects and categorised VM CSFs into four categories, as shown in Table 1 [53]. In addition, three new factors were added to the list and several VM CSFs were modified, as shown in Table 1 [53]. The revised and new tasks were utilised to create the pilot study questionnaire.

**Table 1.** VM CSFs.

| CSFs Subscales (Groups) | Code | Name | Researches |
|---|---|---|---|
| | SF.SK1 | Constructing a VM team from a variety of discipline | [15,17,23] |
| | SF.SK2 | Competence of VM facilitator | [15,23,37,59, 60] |
| | SF.SK3 | Collaborative discussion that is well-communicated | [61,62] |
| | SF.SK4 | Capability to lead a VM workshop | [27] |
| | SF.SK5 | Ability to use and learn about VM | [15,17,23] |
| Knowledge and Stakeholders | SF.SK6 | Participation of all relevant parties in the VM workshop | [17,61] |
| | SF.SK7 | Professional knowledge and expertise in the subjects of the participants | [17] |
| | SF.SK8 | Readiness to embrace novel ideas and approaches | [17] |
| | SF.SK9 | Establishing the roles and purposes of various professions | [61] |
| | SF.SK10 | Consumer involvement | [23] |
| | SF.SK11 | Competence and character traits of the individuals involved | [17] |
| | SF.SK12 | Stakeholder and agency cooperation and a high-quality working relationship | [17,59,62] |
| | SF.SK13 | Participant discipline and attitude | [23] |
| | SF.CE1 | Workshop attendees articulated their VM's clear and defined purpose | [23,62] |
| Environment and Culture | SF.CE2 | Participant organisations' delegation of decision-making authority | [23] |
| | SF.CE3 | Identifying and articulating the core values of a target audience | [63] |
| | SF.CE4 | Motivate VM designer to generate VM outputs | [64] |
| | SF.WD1 | A proactive, imaginative, and organised strategy | [15,17] |
| | SF.WD2 | Function and component analysis of the project | [17] |
| | SF.WD3 | VM feedback mechanism | [27] |
| | SF.WD4 | Customers' understanding of VM's value-optimization potential | [27] |
| | SF.WD5 | Appropriate input from the original design team | [60] |
| Dynamics of the Workshop | SF.WD6 | VM workshop was appropriately timed. | [17] |
| | SF.WD7 | Gathering of contextual data | [23] |
| | SF.WD8 | Group orientation | [62,65] |
| | SF.WD9 | Innovative method of generating ideas | [64] |
| | SF.WD10 | Improved rates of innovation and assessment through the use of cutting-edge technology | [64] |
| | SF.WD11 | Integration of virtual reality workshops into the project lifecycle | [23] |
| | SF.ST1 | Clients' involvement and encouragement | [15,17,23,62] |
| | SF.ST2 | Suggestions from the proper state and municipal agencies | [37] |
| Standardisation | SF.ST3 | Consistent presence of the policy maker | [15] |
| | SF.ST4 | VM workshop strategy for execution | [15,17,37] |
| | SF.ST5 | An official government promise to adopt VM | [6] |

### 3.2. Pilot Survey

A pilot study in the Egyptian residential construction industry was undertaken to explore the results mentioned above through EFA, which sent the pilot questionnaire to an appropriate number of participants (200 construction professionals) [53,66]. The research instrument's reliability was tested using the Cronbach alpha test. This test enables assessing the reliability of the in-area questionnaire and all the fields considered. The alpha values obtained ranged from 0.84 to 0.91, indicating a high-reliability level for the study's surveys [53,67].

### 3.3. Main Survey

As VM implementation is relatively new in Egypt, a stratified sampling of the specific subspecies has been considered [23]. Stratification considers demographic variations in the three industries (client, consultant, and contractor) [68]. The screening study created over 280 entities, although the proposal was only supported by 215. The survey was used to assess the level of VM implementation, awareness, and to identify essential CSFs using the research instrument (Questionnaire) recommended by Fellows and Liu [69]. Consequently, the results show that the participants have enough VM awareness and knowledge.

### 3.4. Pearson Correlation Analysis

The aim of this study is to identify the critical VM activities in the Egyptian construction industry. Consequently, it is essential to check the correlations between new data from the primary survey. In the natural sciences, the Pearson correlation factor is widely used [70]. It is used to calculate the correlation among two variables X and Y, whose estimates are between −1 and 1 and calculated by the following equation:

$$r = \frac{\sum_{i=1}^{n}\left(X_i - \overline{X}\right)\left(Y_i - \overline{Y}\right)}{\sqrt{\sum_{i=1}^{n}\left(X_i - \overline{X}\right)^2}\sqrt{\sum_{i=1}^{n}\left(Y_i - \overline{Y}\right)^2}} \tag{1}$$

where $\overline{X}$ = mean value of sample one; $\overline{Y}$ = mean value of sample two; and *r* represents the Pearson correlation coefficient. The estimated value range of r is from −1 to 1. The greater the absolute value, the greater the degree of correlation. The higher the coefficient of correlation is to 1 or −1, the greater the degree of correlation. Conversely, the quieter the coefficient of correlation to 0, and the lower the correlation. This was used to explain the association between VM CSFs groups of the Pearson coefficient. The correlation of these groups was determined automatically using the Statistical Package for the Social Sciences (SPSS) software.

### 3.5. Ranking Analysis

Relative Importance Index (*RII*) is the most commonly used method for rankings of the attributes [71,72] as identified by Salleh [73] and is a statistical method used to identify ranks of different causes. The response events' frequency and intensity were evaluated in Equation (1) [74,75], using 5-point Likert scale and *RII*.

$$RII = \frac{\sum w}{A \times N} = \frac{5n_5 + 4n_4 + 3n_3 + 2n_2 + 1n_1}{5 \times N}. \tag{2}$$

where *W* indicates the respondent's weighting to each variable, *A* is the maximum weight, and *N* is the whole number of members. Table 2 shows the results of *RII* ranks. This calculation was further be classified using the three selected respondents' groups (owner, consultant, and contractor) to cross-compare the relative significance of the factors perceived by the selected three groups. Using this assessment, the study can identify the most critical VM CSFs contributing to VM implementation in Egypt's residential building industry.

**Table 2.** Pearson Correlation Analysis for Success Factors of VM Implementation.

| VM CSFs Subscales (Groups) | Knowledge/ Stakeholders | Environment and Culture | Dynamics of Workshop | Standardisation |
|---|---|---|---|---|
| Knowledge and Stakeholders | 1 | 0.328 0.000 | 0.236 0.001 | 0.224 0.001 |
| Environment and culture | | 1 | 0.286 0.000 | 0.200 0.003 |
| Dynamics of Workshop | | | 1 | 0.328 0.000 |
| Standardisation | | | | 1 |
| Mean | 3.706 | 3.655 | 3.606 | 3.569 |
| SD | 0.807 | 0.935 | 0.842 | 0.808 |

*3.6. Stationary Analysis (Ginni's Mean)*

To determine the VM CSFs, our study followed Samuel and Ovie's [76] strategy. The following are the steps involved in this strategy:

(a)    As stated in Equation (3), "Ginni's mean difference measure of dispersion" [77] may be used to calculate the average spread of the *RII* values.

$$G.M = \frac{G}{M} \quad (3)$$

Ginni's mean difference (*G.M*) is a measure of dispersion where *N* is the number of factors and *G* is the sum of the changes in value between all imaginable pairs of variables, and *M* is the total number of variances.

$$M = \frac{N(N-1)}{2}. \quad (4)$$

(b)    Equation (5) is used to calculate weights for each *RII* number based on the predicted Ginni's mean difference measure of dispersion:

$$Wi = G.M \times \frac{RIIi}{RII1}. \quad (5)$$

where *RIIi* is the relative index number of any CSFs, *RII*1 is the greatest relative index number, and *Wi* is the weight of each *RII* number.

(c)    RII central value can be represented by approving the geometric mean (*G:M. (w)*) of the RII numbers and by fitting this value to the *RII* calibration to reflect the stationary, as defined by Equation (6):

$$G:M.(w) = Antilog \frac{\sum w.logRII}{\sum w}. \quad (6)$$

where $\sum w$: is equal to the weights given to the *RII* numbers as a whole.

## 4. Data Analysis

*4.1. Pearson Correlation Analysis*

The Pearson association between the VM CSFs was analysed using SPSS software. The results are indicated in the descriptive and correlation results presented in Table 2. For Stakeholders and Knowledge (M = 3.706, SD = 0.807), for Culture and Environment (M = 3.655, SD = 0.935), for Workshop Dynamics (M = 3.606, SD = 0.842), and finally for Standardisation (M = 3.569, SD = 0.808). The Pearson correlation gives both directions (positive or negative) and the strength of a relationship between two variables [78]. A positive association and correlation indicate that if one variable increases, the other variable also rises. However, a negative association and correlation mean that if one variable

increases, the other variable reduces [79,80]. The Pearson correlation coefficient fluctuates from −1 (highly negative correlation) to +1 (highly positive correlation). Coefficients of correlation suggest a significant positive correlation among groups of VM CSFs.

### 4.2. Critical VM CSFs in the Residential Building Industry

Several VM CSFs have been identified, leading to considerable increases in the success of VM deployment in residential building projects. The research identified 34 VM CSFs that could be used to implement VM (Table 1). The data obtained from the questionnaire were submitted to the SPSS software and analysed using the *RII* to examine the relative importance of CSFs affecting VM implementation. The *RII* number is from −1 to 1, with −1, not an option. To evaluate the *RII* against the appropriate significance level, we apply the transformation matrix proposed by Chen et al. [81]. Table 3 displays the cut-offs based on the RII importance.

**Table 3.** RII Importance levels.

| Rank of Significance | Range |
|---|---|
| High (H) | 0.8 < RII < 1.0 |
| High-Medium (H-M) | 0.6 < RII < 0.8 |
| Medium (M) | 0.4 < RII < 0.6 |
| Medium-Low (M-L) | 0.2 < RII < 0.4 |
| Low (L) | 0.0 < RII < 0.2 |

Table 4 summarises the RII results of CSFs of VM implementation. Stakeholders and Knowledge is the largest group with 13 items, followed by Culture and Environment with 4, Workshop Dynamics with 11, and Standardisations at 6. Each rating system must use a 5-point Likert scale. Figure 2 and Table 4 showed the RII results of the VM CSFs, along with the corresponding ranking and significance level. All factors have been assigned a "High-Medium" relevance level, except for the three "High" factors (SF.SK5, SF.SK7, and SF.WD7), as revealed by the ranking results. However, the top five ranking factors across all participants with the highest RII above 0.75 are: constructing a VM team from a variety of disciplines, participants' awareness of VM, professional experience of the respective disciplines of the participants, and collected background information. Most of these factors target professionals through the implementation of VM. The RII of VM CSFs and the standard deviation (SD) from the mean demonstrate that the respondents' perceptions were very satisfactory and notably distinctive.

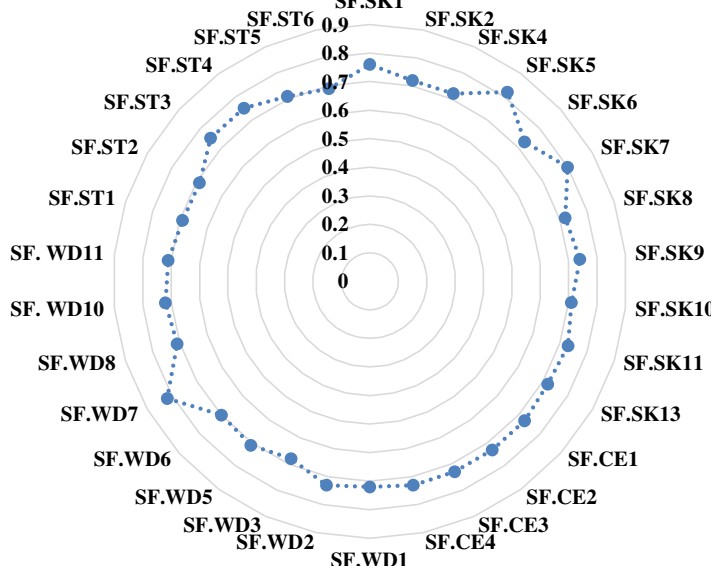

**Figure 2.** RII levels for VM CSFs.

**Table 4.** Descriptive Numbers Concerning Veterinary CSFs.

| VM CSFs Subscales (Groups) | Item | RII | SD | Level of Importance |
|---|---|---|---|---|
| Knowledge/Stakeholders | SF.SK1 | 0.76 | 1.19 | H-M |
| | SF.SK2 | 0.72 | 0.99 | H-M |
| | SF.SK4 | 0.71 | 1.08 | H-M |
| | SF.SK5 | 0.83 | 1.09 | H |
| | SF.SK6 | 0.7 | 1.20 | H-M |
| | SF.SK7 | 0.8 | 1.17 | H |
| | SF.SK8 | 0.71 | 1.29 | H-M |
| | SF.SK9 | 0.739 | 1.25 | H-M |
| | SF.SK10 | 0.71 | 1.24 | H-M |
| | SF.SK11 | 0.722 | 1.19 | H-M |
| | SF.SK13 | 0.72 | 1.21 | H-M |
| | Total | 0.73 | 0.82 | H-M |
| Environment and Culture | SF.CE1 | 0.72 | 1.04 | H-M |
| | SF.CE2 | 0.719 | 1.04 | H-M |
| | SF.CE3 | 0.725 | 1.06 | H-M |
| | SF.CE4 | 0.715 | 1.10 | H-M |
| | Total | 0.73 | 0.93 | H-M |
| Dynamics of Workshop | SF.WD1 | 0.72 | 0.94 | H-M |
| | SF.WD2 | 0.725 | 1.03 | H-M |
| | SF.WD3 | 0.68 | 1.10 | H-M |
| | SF.WD5 | 0.71 | 1.07 | H-M |
| | SF.WD6 | 0.7 | 0.99 | H-M |
| | SF.WD7 | 0.82 | 1.03 | H |
| | SF.WD8 | 0.7 | 0.94 | H-M |
| | SF.WD10 | 0.74 | 1.20 | H-M |
| | SF.WD11 | 0.71 | 1.06 | H-M |
| | Total | 0.72 | 0.85 | H-M |
| Standardization | SF.ST1 | 0.69 | 1.00 | H-M |
| | SF.ST2 | 0.69 | 1.03 | H-M |
| | SF.ST3 | 0.72 | 1.07 | H-M |
| | SF.ST4 | 0.75 | 1.09 | H-M |
| | SF.ST5 | 0.7 | 0.89 | H-M |
| | SF.ST6 | 0.69 | 1.00 | H-M |
| | Total | 0.71 | 0.81 | H-M |

*4.3. Stationary Value Management Implementation CSFs*

Ginni's coefficient of mean difference can be found using Equation (4) and Table 5, which enable understanding the RII number for each criterion. To get Ginni's coefficient of mean difference (G.M), we must first tally the differences in value for all possible pairings of independent variables. Table 5 displays the RII differences between all couples. The sum of all possible value differences between all possible pairs of variables is G = 15.6, as shown in Table 6. The table also presented the total number of variations between all pairs of variables as 435, as determined using the following Equation (5). Table 6 showed that the weighted geometric mean G.M. (w) is 0.735, due to the relationships between the parameters $\sum w = 0.94$ and $\sum w$. Log RII = $-0.1259$. The RII value obtained is consistent with the RII data for SF.SK9 as given in Table 6. Therefore, these characteristics are regarded as the permanent success determinants for construction projects in Egypt.

**Table 5.** Variations of pairs of RII number.

| .Rank | Criterion | RII | 1 | 2 | 3 | 4 | 5 | 6 | 7 | 8 | 9 | 10 | 11 | 12 | 13 | 14 | 15 | 16 | 17 | 18 | 19 | 20 | 21 | 22 | 23 | 24 | 25 | 26 | 27 | 28 | 29 | Sum |
|---|---|---|---|---|---|---|---|---|---|---|---|---|---|---|---|---|---|---|---|---|---|---|---|---|---|---|---|---|---|---|---|---|
| 1 | SF.SK5 | 0.83 | 0.15 | | | | | | | | | | | | | | | | | | | | | | | | | | | | | 0.15 |
| 2 | SF.WD7 | 0.82 | 0.14 | 0.14 | | | | | | | | | | | | | | | | | | | | | | | | | | | | 0.28 |
| 3 | SF.SK7 | 0.8 | 0.14 | 0.13 | 0.12 | | | | | | | | | | | | | | | | | | | | | | | | | | | 0.39 |
| 4 | SF.SK1 | 0.76 | 0.14 | 0.13 | 0.11 | 0.08 | | | | | | | | | | | | | | | | | | | | | | | | | | 0.46 |
| 5 | SF.ST4 | 0.75 | 0.13 | 0.13 | 0.11 | 0.07 | 0.070 | | | | | | | | | | | | | | | | | | | | | | | | | 0.51 |
| 6 | SF.WD10 | 0.74 | 0.13 | 0.12 | 0.11 | 0.07 | 0.060 | 0.060 | | | | | | | | | | | | | | | | | | | | | | | | 0.55 |
| 7 | SF.SK9 | 0.739 | 0.13 | 0.12 | 0.10 | 0.07 | 0.060 | 0.050 | 0.059 | | | | | | | | | | | | | | | | | | | | | | | 0.59 |
| 8 | SF.CE3 | 0.725 | 0.13 | 0.12 | 0.10 | 0.06 | 0.060 | 0.050 | 0.049 | 0.045 | | | | | | | | | | | | | | | | | | | | | | 0.61 |
| 9 | SF.WD2 | 0.725 | 0.12 | 0.12 | 0.10 | 0.06 | 0.050 | 0.050 | 0.049 | 0.035 | 0.045 | | | | | | | | | | | | | | | | | | | | | 0.63 |
| 10 | SF.SK11 | 0.722 | 0.12 | 0.11 | 0.10 | 0.06 | 0.050 | 0.040 | 0.049 | 0.035 | 0.035 | 0.042 | | | | | | | | | | | | | | | | | | | | 0.64 |
| 11 | SF.SK2 | 0.72 | 0.12 | 0.11 | 0.09 | 0.06 | 0.050 | 0.040 | 0.039 | 0.035 | 0.035 | 0.032 | 0.040 | | | | | | | | | | | | | | | | | | | 0.65 |
| 12 | SF.SK13 | 0.72 | 0.12 | 0.11 | 0.09 | 0.05 | 0.050 | 0.040 | 0.039 | 0.025 | 0.035 | 0.032 | 0.030 | 0.040 | | | | | | | | | | | | | | | | | | 0.66 |
| 13 | SF.CE1 | 0.72 | 0.12 | 0.11 | 0.09 | 0.05 | 0.040 | 0.040 | 0.039 | 0.025 | 0.025 | 0.032 | 0.030 | 0.030 | 0.040 | | | | | | | | | | | | | | | | | 0.67 |
| 14 | SF.WD1 | 0.72 | 0.12 | 0.11 | 0.09 | 0.05 | 0.040 | 0.030 | 0.039 | 0.025 | 0.025 | 0.022 | 0.030 | 0.030 | 0.030 | 0.040 | | | | | | | | | | | | | | | | 0.68 |
| 15 | SF.ST3 | 0.720 | 0.11 | 0.11 | 0.09 | 0.05 | 0.040 | 0.030 | 0.029 | 0.025 | 0.025 | 0.022 | 0.020 | 0.030 | 0.030 | 0.030 | 0.040 | | | | | | | | | | | | | | | 0.68 |
| 16 | SF.CE2 | 0.719 | 0.11 | 0.10 | 0.09 | 0.05 | 0.040 | 0.030 | 0.029 | 0.015 | 0.025 | 0.022 | 0.020 | 0.020 | 0.030 | 0.030 | 0.030 | 0.039 | | | | | | | | | | | | | | 0.68 |
| 17 | SF.CE4 | 0.715 | 0.11 | 0.10 | 0.08 | 0.05 | 0.040 | 0.030 | 0.029 | 0.015 | 0.015 | 0.022 | 0.020 | 0.020 | 0.020 | 0.030 | 0.030 | 0.029 | 0.035 | | | | | | | | | | | | | 0.67 |
| 18 | SF.SK4 | 0.71 | 0.11 | 0.10 | 0.08 | 0.04 | 0.035 | 0.030 | 0.029 | 0.015 | 0.015 | 0.012 | 0.020 | 0.020 | 0.020 | 0.020 | 0.030 | 0.029 | 0.025 | 0.030 | | | | | | | | | | | | 0.66 |
| 19 | SF.SK8 | 0.71 | 0.11 | 0.10 | 0.08 | 0.04 | 0.031 | 0.025 | 0.029 | 0.015 | 0.015 | 0.012 | 0.010 | 0.020 | 0.020 | 0.020 | 0.020 | 0.029 | 0.025 | 0.020 | 0.030 | | | | | | | | | | | 0.65 |
| 20 | SF.SK10 | 0.71 | 0.11 | 0.10 | 0.08 | 0.04 | 0.030 | 0.021 | 0.024 | 0.015 | 0.015 | 0.012 | 0.010 | 0.010 | 0.020 | 0.020 | 0.020 | 0.019 | 0.025 | 0.020 | 0.020 | 0.020 | | | | | | | | | | 0.63 |
| 21 | SF.WD5 | 0.71 | 0.11 | 0.10 | 0.08 | 0.04 | 0.030 | 0.020 | 0.020 | 0.010 | 0.015 | 0.012 | 0.010 | 0.010 | 0.010 | 0.020 | 0.020 | 0.019 | 0.015 | 0.020 | 0.020 | 0.020 | 0.030 | | | | | | | | | 0.63 |
| 22 | SF.WD11 | 0.71 | 0.11 | 0.10 | 0.08 | 0.04 | 0.030 | 0.020 | 0.019 | 0.006 | 0.010 | 0.012 | 0.010 | 0.010 | 0.010 | 0.010 | 0.020 | 0.019 | 0.015 | 0.010 | 0.020 | 0.020 | 0.020 | 0.030 | | | | | | | | 0.61 |
| 23 | SF.SK6 | 0.7 | 0.11 | 0.10 | 0.08 | 0.04 | 0.030 | 0.020 | 0.019 | 0.005 | 0.006 | 0.007 | 0.010 | 0.010 | 0.010 | 0.010 | 0.010 | 0.019 | 0.015 | 0.010 | 0.010 | 0.020 | 0.020 | 0.020 | 0.020 | | | | | | | 0.59 |
| 24 | SF.WD6 | 0.7 | 0.09 | 0.10 | 0.08 | 0.04 | 0.030 | 0.020 | 0.019 | 0.005 | 0.005 | 0.003 | 0.005 | 0.010 | 0.010 | 0.010 | 0.010 | 0.009 | 0.015 | 0.010 | 0.010 | 0.010 | 0.020 | 0.020 | 0.010 | 0.020 | | | | | | 0.55 |
| 25 | SF.WD8 | 0.7 | 0.09 | 0.08 | 0.08 | 0.04 | 0.028 | 0.020 | 0.019 | 0.005 | 0.005 | 0.002 | 0.001 | 0.005 | 0.010 | 0.010 | 0.010 | 0.009 | 0.005 | 0.010 | 0.010 | 0.010 | 0.010 | 0.020 | 0.010 | 0.010 | 0.020 | | | | | 0.51 |
| 26 | SF.ST5 | 0.7 | 0.08 | 0.08 | 0.06 | 0.04 | 0.025 | 0.018 | 0.019 | 0.005 | 0.005 | 0.002 | 0.000 | 0.001 | 0.005 | 0.010 | 0.010 | 0.009 | 0.005 | 0.000 | 0.010 | 0.010 | 0.010 | 0.010 | 0.010 | 0.010 | 0.010 | 0.020 | | | | 0.46 |
| 27 | SF.ST1 | 0.69 | 0.07 | 0.07 | 0.06 | 0.02 | 0.025 | 0.015 | 0.017 | 0.005 | 0.005 | 0.002 | 0.000 | 0.000 | 0.001 | 0.005 | 0.010 | 0.009 | 0.005 | 0.000 | 0.000 | 0.010 | 0.010 | 0.010 | 0.000 | 0.010 | 0.010 | 0.010 | 0.010 | | | 0.39 |
| 28 | SF.ST2 | 0.69 | 0.03 | 0.06 | 0.05 | 0.02 | 0.011 | 0.015 | 0.014 | 0.003 | 0.005 | 0.002 | 0.000 | 0.000 | 0.000 | 0.001 | 0.005 | 0.009 | 0.005 | 0.000 | 0.000 | 0.000 | 0.010 | 0.010 | 0.000 | 0.000 | 0.010 | 0.010 | 0.000 | 0.010 | | 0.28 |
| 29 | SF.ST6 | 0.69 | 0.01 | 0.02 | 0.04 | 0.01 | 0.010 | 0.001 | 0.014 | 0.000 | 0.003 | 0.002 | 0.000 | 0.000 | 0.000 | 0.000 | 0.001 | 0.004 | 0.005 | 0.000 | 0.000 | 0.000 | 0.000 | 0.010 | 0.000 | 0.000 | 0.000 | 0.010 | 0.000 | 0.000 | 0.020 | 0.14 |
| 30 | SF.WD3 | 0.68 | 0.00 | 0.00 | 0.00 | 0.00 | 0.000 | 0.000 | 0.000 | 0.000 | 0.000 | 0.000 | 0.000 | 0.000 | 0.000 | 0.000 | 0.000 | 0.000 | 0.000 | 0.000 | 0.000 | 0.000 | 0.000 | 0.000 | 0.000 | 0.000 | 0.000 | 0.000 | 0.000 | 0.000 | 0.000 | 0.00 |
| Sum | | | 3.16 | 2.87 | 2.31 | 1.23 | 0.965 | 0.715 | 0.691 | 0.369 | 0.369 | 0.306 | 0.266 | 0.266 | 0.266 | 0.266 | 0.266 | 0.251 | 0.195 | 0.130 | 0.130 | 0.120 | 0.130 | 0.130 | 0.050 | 0.050 | 0.050 | 0.050 | 0.010 | 0.010 | 0.020 | 15.60 |

**Table 6.** Estimates of the geometric mean with weights.

| CSFs | RII | Wi | Log RII | Wi. Log RII |
|:---:|:---:|:---:|:---:|:---:|
| SF.SK5 | 0.83 | 0.0359 | −0.0809 | −0.0029 |
| SF.WD7 | 0.82 | 0.0354 | −0.0862 | −0.0031 |
| SF.SK7 | 0.8 | 0.0346 | −0.0969 | −0.0033 |
| SF.SK1 | 0.76 | 0.0328 | −0.1192 | −0.0039 |
| SF.ST4 | 0.75 | 0.0324 | −0.1249 | −0.0040 |
| SF.WD10 | 0.74 | 0.032 | −0.1308 | −0.0042 |
| SF.SK9 | 0.739 | 0.0319 | −0.1314 | −0.0042 |
| SF.CE3 | 0.725 | 0.0313 | −0.1397 | −0.0044 |
| SF.WD2 | 0.725 | 0.0313 | −0.1397 | −0.0044 |
| SF.SK11 | 0.722 | 0.0312 | −0.1415 | −0.0044 |
| SF.SK2 | 0.72 | 0.0311 | −0.1427 | −0.0044 |
| SF.SK13 | 0.72 | 0.0311 | −0.1427 | −0.0044 |
| SF.CE1 | 0.72 | 0.0311 | −0.1427 | −0.0044 |
| SF.WD1 | 0.72 | 0.0311 | −0.1427 | −0.0044 |
| SF.ST3 | 0.72 | 0.0311 | −0.1427 | −0.0044 |
| SF.CE2 | 0.719 | 0.0311 | −0.1433 | −0.0045 |
| SF.CE4 | 0.715 | 0.0309 | −0.1457 | −0.0045 |
| SF.SK4 | 0.71 | 0.0307 | −0.1487 | −0.0046 |
| SF.SK8 | 0.71 | 0.0307 | −0.1487 | −0.0046 |
| SF.SK10 | 0.71 | 0.0307 | −0.1487 | −0.0046 |
| SF.WD5 | 0.71 | 0.0307 | −0.1487 | −0.0046 |
| SF.WD11 | 0.71 | 0.0307 | −0.1487 | −0.0046 |
| SF.SK6 | 0.7 | 0.0302 | −0.1549 | −0.0047 |
| SF.WD6 | 0.7 | 0.0302 | −0.1549 | −0.0047 |
| SF.WD8 | 0.7 | 0.0302 | −0.1549 | −0.0047 |
| SF.ST5 | 0.7 | 0.0302 | −0.1549 | −0.0047 |
| SF.ST1 | 0.69 | 0.0298 | −0.1612 | −0.0048 |
| SF.ST2 | 0.69 | 0.0298 | −0.1612 | −0.0048 |
| SF.ST6 | 0.69 | 0.0298 | −0.1612 | −0.0048 |
| SF.WD3 | 0.68 | 0.0297 | −0.1675 | −0.0050 |
| Sum | | 0.94 | | −0.1259 |

## 5. Discussion

### 5.1. VM CSFs

Pearson correlation analysis was used to validate the results and subscales obtained from the pilot study phase. Consequently, the VM CSFs executed were extracted under "Knowledge and Stakeholders", "Environment and Culture", "Dynamics of Workshop", and "Standardisation". This finding agreed with the finding regarding VM exploring CSFs in the exiting literature. For instance, according to Tanko et al. [82], in the Nigerian construction industry, CSFs can be categorized under "People", "Government", "Environment", and "Information/Methodology". In this study, an analysis of the relative significance (RII) of the extracted groups and their components was performed. This is further discussed below:

### 5.1.1. Knowledge and Stakeholders

The first factor that contributed to the success of VM deployment was connected to "Knowledge and Stakeholders", the RII for "Ability to use and learn about VM" (RII = 0.82, SD = 1.09) and "Professional knowledge and expertise in the subjects of the participants" (RII = 0.80, SD = 1.17) have the maximum level. The bottom mean value belongs to the items "Consumer Involvement" with (RII = 0.71, SD = 1.24) and "Willingness to accept changes and innovations" with (RII = 0.72, SD = 1.29). An RII of 0.738 indicates that all indicators are correspondingly significant. It further indicates that the level of stakeholder and knowledge-related success elements during VM deployment was more significant than the median of the scale (High-Medium level), which is better than the sufficient level. These results concur with Tanko et al. [27]. Their study argued that the factors related to individuals and stakeholders would be crucial to efficiently enable the VM study with the required experience, as VM needs people and stakeholders. It is a proactive and innovative approach. These stakeholders are involved in the dynamic procedure that demands their obligations [83] and an active contribution to achieving the objectives of the workshop [84,85]. Realizing these goals may ultimately aid in the project's development in both direct and indirect ways. Hence, Fong et al. [86] posited that the ever-changing nature of projects in recent years necessitates a collaborative effort from all parties involved to create a project value team with creative solutions and insights.

### 5.1.2. Environment and Culture

"Identifying and articulating the core values of a target audience" ranks highest on the "Culture and Environment" dimension of the CSFs for VM implementation. It has a mean RII of 0.73 and a standard deviation of 1.06. The lowest mean value belongs to "Encourage the VM team to provide VM deliverables" (RII = 0.73, SD = 1.10). Additionally, there is an average relevance of RII = 0.732 across all indicators. It shows that the level of this group during VM deployment was above the median of the scale (high-medium level), showing as more significant than the moderate level for this indicator. This finding is in line with the Tanko et al. [27] who argued that the "environment" component encompasses the innovative ways in which VM members think and the ability to identify, define, and categorise the goals of construction projects through a collaborative and problem-solving approach. The "Culture and the Environment" comprise the situations and environment in which stakeholders operate to enable effective interactions and working relations [37]. Collaboration and cooperation among VM team members will promote innovation and creative alternatives for problems [17,61,62]. Consequently, the organisation's value system will improve, and the client will be more convinced to implement VM [63].

### 5.1.3. Dynamics of Workshop

"Dynamics of workshop" was the subject of the third subscale of the CSFs for the VM implementation. The RII value for "Gathering of Contextual Data" has the highest value with (RII = 0.73, SD = 1.03). The lowest mean value belongs to two items, "Awareness on the part of clients concerning value optimization role of VM" with (RII = 0.7, SD = 1.00) and "VM feedback mechanism" with (RII = 0.68, SD = 1.10). Across all parameters, RII = 0.722 which indicates a high level of significance. It suggests that stakeholder interest and expert knowledge were significantly greater than average during the VM implementation process (high-medium level). Ramly et al. [23] revealed results from several studies indicating that the structured procedures and the work plan constitute VM's core values that distinguish them from other management techniques. Researchers believe each professional must prepare information on how the project relates to them in the VM study [27]. The methodology is a procedure that should be performed to execute a project [27]. Hence, Ramly et al. [23] argued that the VM study's best practice is the commitment of the team leader to facilitate the procedures under the VM work plan. However, the design team can devise numerous ways to enhance the project through workshops [60]. In the same vein, Coetzee [87] posited that technical advancements should be considered and utilized

in VM activities. Using electronic devices to establish VM will save money and time by eliminating the need to bring together physical team members [88]. Moreover, it will enable a virtual VM environment by rewarding professionals with the requisite competencies and skills [89].

5.1.4. Standardization

The "VM study plan for execution" RII (RII = 0.75, SD = 1.09) has the highest score for the "Standardization" subscale of the CSFs of VM implementation. The median value of "The client's tenacity in conveying their demands and constraints to the design team" is 0.69 (RII = 0.69, SD = 1.00). All indicators were given an aggregate significance of RII = 0.7. This indicator is more significant than average because the stakeholder and expertise factor levels for successful VM implementation are above the median value. As it is the major consumer and investor, with much of the capital formation linked to investment in properties and infrastructure, the government should produce all VM's policies [82]. Therefore, it would be crucial to have the government's backing and involvement in imposing the VM on modern building practices [82].

*5.2. The Managerial Contribution of This Study*

Egyptian and stakeholders elsewhere in developing countries could use the VM CSFs from this study as a framework to carry out VM more efficiently to reduce the cost and improve their projects' quality. Consequently, Egypt needs to implement VM to acquire sustainable value since the investment is regularly connected to building development [90]. The proposed VM implementation framework would help Egypt's ambition to have a sustainable economy and become one of the top 30 nations in the world in future [91]. Furthermore, the framework proposed by this study can help encourage the adoption of VM in other developing countries where residential building projects are implemented using similar strategies [92]. This is essential to these countries since they have many financial challenges adapting their building projects [93]. This study can also donate to the exiting body of knowledge by offering three new CSFs which were not highlighted before in the previous VM studies. These might require further analysis of the construction project: "Motivate VM team members to produce VM output", "Innovative method of generating ideas" and "Improved rates of innovation and assessment through the use of cutting-edge technology". Finally, while this study has significantly contributed to both the implication and our body of knowledge, such limitations overture chances for future research. The data analysis depends on two hundred and twelve respondents. Another important influence may be accomplished by a larger sample size. This research portrayed the three types of respondents (owner, consultant, and contractor) as a single, unified entity. Future studies will pursue the development of the relationship using causal and predictive analysis among the various user groups in the building industry.

By rearranging CSFs, stakeholders such as project owners and contractors may produce a "road map" for more efficient VM implementation in their projects. Moreover, this reorganization can serve as a standard for establishing a practical framework for effectively transitioning construction actors across VM phases. This will replace the old environmental and sustainable performance standards that have been in place since the Arab Spring in 2011 [94]. Since the economy is often linked to the argument for sustainable growth [90], Egypt must implement VM if it ever hopes to have a genuinely sustainable economy.

The "road map" will help Egypt achieve its goal of becoming one of the world's top 30 economies and a leading tourist destination [91]. The "road map" created from this research may also be used to stimulate the use of VM in other developing countries concerning sustainable construction projects [92]. This is especially crucial in underdeveloped nations, often hampered by factors like the need to pay astronomical sums of money to address environmental concerns [93]. Consequently, VM could provide these nations with the chance to include eco-friendly practices in their building projects' conceptualization

stages [13,95]. However, the following areas in which this research significantly contributes each have important ramifications for the construction industry:

- To determine whether or not the VM standards can remain competitive and successful in a global market, this database covers the VM standards and the related criteria.
- It helps owners, consultants, and contractors evaluate and pick virtual reality (VR) deployment to improve construction projects' consistency, efficiency, and planning.
- It presents evidence from the scientific community that might help Egypt and other developing nations to embrace VM.
- Virtual reality (VR) and VR research in the field of construction have primarily focused on developed countries, and some few emerging countries (Malaysia, China, and Saudi Arabia, for example). Consequently, there is a lack data concerning VM adoption in developing countries and particularly analysis of VM's use in Egypt's construction sector. Therefore, our study has effectively established a link between VM and the Egyptian construction sector. Hence the study has laid a solid foundation on which to continue discussing the use of VM to increase the dependability of local construction projects and reduce the knowledge gap.
- The results of this study can help advance the use of virtualization technology in construction in Egypt. This study has explained why organizations should adopt virtual machines. Benefits include saving money and ensuring resources are allocated reasonably amongst projects. Thus, by designing and implementing the intended methods, all stakeholders may concentrate on the project's goal in terms of cost, timeliness, and efficacy. Achieving a high level of sustainability in a project has good consequences in the long run.
- The findings of this study can also be used as a standard to measure the success of a project's execution and identify areas where improvements can be made. Issues including budget overruns, completing projects on time, and vague requirements all fell under this category. Additionally, business owners and managers could have access to information from this study that will help them implement VM in ways that increase the success of their enterprises.
- Critical success factors (CSFs) for VM deployment were found, as were other VM-related criteria not included in earlier research.

### 5.3. Theoretical Implications

Although research on sustainable concepts is not new [96], it is becoming increasingly important to many businesses [97]. The suggested prioritization approach stipulates the need for VM deployment, particularly around green home construction. Through the suggested methodology, this research discovered the CSFs for VM deployment. The challenges of deploying VM in Egypt's construction sector may be overcome with the help of these CSFs. Also, this research will help bridge the gap between VM theory and practice. However, as far as we are aware, no research has been done to look at the CSFs of VM deployment in the Egyptian construction industry. In the first place, this research provides an empirical identification of the crucial CSFs of VM that might facilitate the adoption of VM in the building sector. This discovery lays the groundwork for scholars, especially those in construction management, to investigate the CSFs of VM in underdeveloped nations.

### 6. Conclusions

This research has demonstrated our new efforts concerning exploring the critical success factors of value management implementation for sustainable residential building projects using a stationary analysis approach. The main contribution of this research paper was that it assessed VM CSFs, which were derived from the literature. Therefore, the validation of the VM CSFs for major groups was carried out through a semi-structured interview and the EFA analysis. A questionnaire survey and subsequent statistical analysis were used to identify the CSFs empirically. This study discovered that the CSFs could benefit from VM's potential applications in the Egyptian construction industry and other

developing nations with similar settings. The careful consideration of the CSFs during VM workshops' initial planning stages increases workshop performance and incentives for completing the project. The results will, in practice, help identify possible deficiency areas such that effective standard corrective action based on VM attributes can be proactively taken. The study has offered insights into current theories, including issues in the area of VM practice in academics and management. To this end, the study's hypothesis has been accepted because all participants showed that VM CSFs are vital to be implemented in their projects. The study outlined a set of guidelines for future researchers to follow, as well as a different VM framework that professionals in the field may use to ensure VM is being implemented and total quality products are being made.

**Author Contributions:** Conceptualization, A.F.K.; methodology, A.F.K.; software A.F.K.; validation, A.F.K.; formal analysis, A.F.K.; investigation, A.F.K.; resources, A.F.K.; data curation, A.F.K.; writing—original draft preparation, A.F.K.; writing—review and editing, A.F.K.; visualization, A.F.K., M.S.U. and A.F.M.; supervision, A.F.K.; project administration, A.F.K.; content, A.F.K. All authors have read and agreed to the published version of the manuscript.

**Funding:** The authors extend their appreciation to the Deputyship for Research and Innovation, Ministry of Education in Saudi Arabia, for funding this research work through the project number (IF2/PSAU/2022/01/22700).

**Acknowledgments:** The authors extend their appreciation to the Deputyship for Research and Innovation, Ministry of Education in Saudi Arabia, for funding this research work through the project number (IF2/PSAU/2022/01/22700).

**Conflicts of Interest:** The authors declare no conflict of interest.

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
