# Peer review of "Exploring the Critical Success Factors of Value Management Implementation for Sustainable Residential Building Project: A Stationary Analysis Approach"

_sustainability, doi:10.3390/su142316215_

Round 1

Reviewer 1 Report

"Identifying and Assessing Critical Success Factors of Value Management Implementation for Sustainable Residential Building Project: A Stationary Analysis Approach" The article is interesting. A few observations are given below,

1) The abstract is not clear. An abstract is a short summary of your completed research. It is intended to describe your work without going into great detail. Abstracts should be self-contained and concise, explaining your work as briefly and clearly as possible. Please take some time to revise and rearrange the abstract to highlight the overall of your research. 

2) For readers to quickly catch the contribution of this work, it would be better to highlight major difficulties and challenges, and the authors' original achievements to overcome them, in a clearer way in the Introduction section. Also, the recent literature available in this field should be included in the introduction section. 

Author Response

"Identifying and Assessing Critical Success Factors of Value Management Implementation for Sustainable Residential Building Project: A Stationary Analysis Approach" The article is interesting. A few observations are given below,

Thank you very much for reviewing our manuscript. We also greatly appreciate the reviewers for their complimentary comments and suggestions. We have carried out the modifications that the reviewer suggested and revised the manuscript accordingly in the word file. Please find below a point-by-point response to the reviewers’ concerns. We hope that you find our responses satisfactory and that the manuscript is now acceptable for publication.

1) The abstract is not clear. An abstract is a short summary of your completed research. It is intended to describe your work without going into great detail. Abstracts should be self-contained and concise, explaining your work as briefly and clearly as possible. Please take some time to revise and rearrange the abstract to highlight the overall of your research.

Thank you for your feedback, we agree with the reviewer. The abstract has been refined and highlighted the whole research work. Kindly refer to the abstract section:

During the past two decades, value management (VM), has developed into a recognized construction practice. However, the methods and activities associated with VM adopt informal approaches in developing countries. This study aims to explore the Critical Success Factors (CSFs) of VM implementation. Consequently, VM CSFs were investigated from previous literature and further categorized over a semi-structured interview. A total of 335 experts in the residential construction industry in Cairo and Giza filled out the questionnaire, and 200 had completed the pilot questionnaire. Subsequently, the exploratory study using the Exploratory Pearson correlation of the VM CSFs was employed to validate the categorization resulting from a semi-structured interview and pilot study phases. Based on the validation results, the VM CSFs may be divided into four dimensions: culture and environment, workshop dynamics, stakeholder and knowledge, and standardization. Through Important Relative Index (RII) analysis, the essential CSFs are Creating a VM Team from a Variety of Discipline, VM knowledge, experience of participants, and professional experience of the different participants' diverse disciplines. In addition, this research used a stationary analytic strategy to evaluate the degree to which VM Critical Success Factors (CSFs) have been incorporated into residential construction projects in Egypt. The results revealed that “Establishing the Roles and Purposes of Various Professions” were the stationary success factor for adopting VM. This research establishes a road map for successful VM implementation via VM CSFs in Egypt and other underdeveloped nations. Stakeholders in the residential construction sector would benefit from this study by learning more about VM CSFs and how they may be used to increase the value of their projects

2) For readers to quickly catch the contribution of this work, it would be better to highlight major difficulties and challenges, and the authors' original achievements to overcome them, in a clearer way in the Introduction section. Also, the recent literature available in this field should be included in the introduction section.

We agree with the reviewer. The whole introduction has been enhanced based on the reviewer’s valuable comments to highlight the major difficulties and challenges in the building industry and how can VM can solve these challenges. In addition, recent studies related to our work have been added based on the reviewer’s comments. Kindly refer to the introduction section:

“Introduction

The residential building market is one of the most dynamic sectors in several nations. The success of residential construction projects takes a back seat to other factors in developing countries. Even though these countries have experienced rapid economic expansion, it is undeniable that the residential construction business is essential to guarantee minimum living standards for the citizens [1]. The residential construction industry has experienced radical change in many developing nations to support their economic goals [2]. It has been established that financial plans in underdeveloped nations are mostly silent during the upgrading phase [3]. Consequently, Ofori [4] noted that poor project management has led to many difficulties for construction companies operating in underdeveloped countries. The project timetable is often not met, and the final budget is much more than projected [5]. In the same vein, Kim, et al. [6] acknowledged that residential development projects were suspended or abandoned because of a lack of finance. Furthermore, there has not been enough efforts to maximise investment in urban housing sectors through residential development projects in developing countries [7].

Egypt is a developing country with extreme-risk markets because of low incomes, high rate of unemployment, and security concerns [8]. Dire currency instability, a dearth of informed commercial opportunities, and the limitations of financial models all contributed to the risk [9]. Despite tremendous progress and rapid population growth between 1950 and 2020, it remained one of the largest inhabited countries in North Africa [10]. Estimates showed that by 2020, the population will be more than five times what it was in 1950. Subsequently, Egyptian policymakers are facing many challenges in achieving their residential building project requirements [11]. This challenge puts difficulty on the Egyptian government to produce sufficient residential building projects, in response to demands by increasing population [12].

Furthermore, despite the increasing population by 9% between 2008 and 2013, the increase in rural development between 2001 and 2012 boosted only by 0.9% [11]. It has highlighted the need to improve "Residential building's success" by adding value, cutting costs, and increasing quality to meet the needs of Egyptian homeowners. Value management (VM) can dramatically integrate the successful approach for residential building projects [13], and VM is recommended as a mechanism for improving the success value of a project [14].

Value management is a multidisciplinary, team-oriented, organisational, and universal analytic tool exclusively designed to help clients achieve their objectives [15, 16]. It recommends a technique that starts with the planning phase, ends the project, stimulates and reduces unnecessary costs [17]. Requests are handled in the construction industry to increase productivity and measures necessary to decrease the residential building project expense. Implementation of VM is supposed to be valuable for project clients, consultants, and contractors [18]. VM has been shown to reduce the cost of investment in construction projects by 10 to 25%, as reported by Ellis, et al. [19]. Similarly, Tang and Bittner [20] argued that VM could systematically explore consistent interpretations of the functions required for the various tasks of marine construction. Besides, over $43,000,000 and 12 months were reduced and saved through the application of VM. This saving enabled the construction industry to realize 6% financial savings and a 17% decline in working time for the building of motorways [21]. However, VM methods do not receive the needed attention in most developing countries, including Egypt [6]. Oke and Ogunsemi [22] establish that insignificant VM studies are continuing, and well-organised workshops are in progress in these countries. Even though many previous studies have covered the advantages, activities, and technological efficacy of VM in many other nations, no attempthas been made to quantify the extent to which VM is used in Egyptian building projects. Keeping in mind that there is a lack of study in this area is essential. This is also true of the Egyptian building sector. Abdelghany, et al. [23] arghued that there were no comprehensive studies conducted to determine the current state of VM education and adoption in Egypt. In Egypt, it is impossible to use the conventional VM approach. Consequently, Othman, et al. [24] reported that the vast majority (86.4%) of construction industry experts do not use virtual reality (VR) in their projects. Of course, this motivates impromptu approaches like disorganised teamwork which donot help in keeping construction expenses down.

The implementation of VM is necessary to control the performance and success of residential building projects., Likewise, the critical success factors (CSFs) are crucial for implementing VM [25]. Research and analysis of this subject were invented by Romani [26]. However, Shen and Liu [17] studied CSFs by choosing various applications in the United Kingdom, Hong Kong, and the United States. Nevertheless, no data have been identified and collected on this issue from the Egyptian building industry's perception. It is argued by Pasquire and Mauro [27] and Hunter and Kelly [28] that modifications in the political, economic, cultural, and project-delivery systems could lead to different CSFs in various geographical regions for the same industry, and CSFs vary from one country to another [29]. Based on these revelations, this study hypothesised that there is a consensus on the significance of VM CSFs in residential construction projects. Hencethis study was motivated by finding answers to the following research question. What are the critical VM CSFs? Consequently, this research tried to answer this question within the context of Egypt through analysing the VM workshops CSFs. Therefore, this method can be revolutionary in the realm of residential construction, especially in underdeveloped nations.

Reviewer 2 Report

The topic of the article is the study of the Value Management Critical Success Factor for Egyptian residential construction projects, and it is well explained in the abstract. The title is rather generic and should better explain the Egyptian focus.

Abstract

The abstract is well structured, but it could present the research with less detail. Data as the number of experts or pilot questionnaire should be presented in the main body text rather than in the abstract.

Introduction

Lines 73-78 introduce the topic of VM for maritime construction, maybe this data is not relevant for Egyptian residential construction. Moreover, it is not clear if the sentence beginning on line 75 is linked to the previous one on maritime construction or with the one before.

Please avoid exclamation marks ad the one between brackets on line 88.

On line 89 virtual reality (VR) is mentioned for the first time, but it seems disconnected from what said until this point. A better introduction and explanation of the benefits of VR could help the reader to better follow the text logic.

Sentence on lines 100-101 talks about consensus on the definition of VM CSFs in residential construction projects. This sentence comes after the suggestion that CSFs fluctuate from one country to another (lines 99-100). Please better state how these conflicting statements are linked.

On line 104 you use the term "revolutionary". Since the research is site specific maybe it would be better to tone down this affirmation.

VM and the sustainable construction industry

Lines 122-130 are not clear from a reader's point of view. Maybe a general rephrasing of the sentences would help in an easier reading. E.g., declare which is the latter factor mentioned on line 125 and state that "these phases" (line 129) are all the six previously mentioned and not only the three made explicit.

On line 166 data about UK are introduced. Since CSFs change depending on the Country, are these data necessary? In the same sentence, on line 167, "VM" is used to shorten "virtual machine" but it can create misunderstanding. Please consider of keeping "virtual machine management" without "(VM)".

Methodology

On line 192 you write "The Pearson correlation analysis was invited". Is "invited" the best term to use?

Please check consistency of CSFs Subscales names between Table 1 and Table 2. Maybe Table 1 should be better placed between section 3.1 and 3.2, since it refers to section 3.1 Semi-structured interviews.

Perhaps Figure 1 could supplement a portion of text explaining the method at the end of the part of section, rather than having the function of explaining the research method.

In section 3.1 the interviews process is explained. The ten interviews mentioned on line 203 were each undertaken by each of the fifteen experts? What about the 355 experts mentioned in the abstract?

On line 216 you mention that three new factors were added. Which factors were added? Base on what were these new factors added?

Are references on line 222 correctly placed?

On line 231 you stated that 230 entities were created, but the proposal was only supported by 215: why?

Check formatting of formula (1).

The equation after line 276 is numbered (5) as the one before. Please check correct numbering.

Data analysis

Check section title formatting on line 278.

On line 282 "Table VII" is mentioned. Is it "Table 2"? If no, please check that "Table 2" is correctly mentioned in-text.

On line 295, 34 VM CSFs are mentioned. Does it refer to Table 1? If yes, maybe mention it since the total number was not mentioned in section 3.1 Semi-structured interviews.

On line 306 there is a reference to "Table VI".

Table 3, on line 333, should be Table 5. Check the formatting since part of the table is missing. Same for the following table.

Discussion

On line 348 the sentence ends with colons followed by sub-sections. Maybe you can complete the sentence after colons to introduce the logic of the following sub-sections.

Conclusions

On line 441 the analysis of 214 samples is mentioned, but this number of samples was not previously mentioned. Maybe can integrate it before in the appropriate section.

Managerial implications

The presence of sections after the Conclusion is quite uncommon. Maybe this section could integrate the discussion section, and the new title for the section could be "5. Results and discussion".

Theoretical implications

As per the previous section, it is quite odd to have other separate sections after the conclusions. Maybe this section can be part of the "6. Conclusion". Usually there is no presence of references at this point of papers, please avoid using them here.

General comments

Some steps should be better explained in the methodology to allow the replication in other research.

Please check if the tables are correctly numbered and mentioned in-text.

References are fairly recent, almost 1/3 is from the past 5 years. Please check for consistency in the used style.

Author Response

Abstract

The abstract is well structured, but it could present the research with less detail. Data as the number of experts or pilot questionnaire should be presented in the main body text rather than in the abstract.

We have carried out the modifications that the reviewer suggested and revised the manuscript accordingly in the word file. Please find below a point-by-point response to the reviewers’ concerns. We hope that you find our responses satisfactory and that the manuscript is now acceptable for publication.

Thank you for your feedback, we agree with the reviewer. The abstract has been refined and highlighted the whole research work. Kindly refer to the abstract section:

During the past two decades, value management (VM), has developed into a recognized construction practice. However, the methods and activities associated with VM adopt informal approaches in developing countries. This study aims to explore the Critical Success Factors (CSFs) of VM implementation. Consequently, VM CSFs were investigated from previous literature and further categorized over a semi-structured interview. A total of 335 experts in the residential construction industry in Cairo and Giza filled out the questionnaire, and 200 had completed the pilot questionnaire. Subsequently, the exploratory study using the Exploratory Pearson correlation of the VM CSFs was employed to validate the categorization resulting from a semi-structured interview and pilot study phases. Based on the validation results, the VM CSFs may be divided into four dimensions: culture and environment, workshop dynamics, stakeholder and knowledge, and standardization. Through Important Relative Index (RII) analysis, the essential CSFs are Creating a VM Team from a Variety of Discipline, VM knowledge, experience of participants, and professional experience of the different participants' diverse disciplines. In addition, this research used a stationary analytic strategy to evaluate the degree to which VM Critical Success Factors (CSFs) have been incorporated into residential construction projects in Egypt. The results revealed that “Establishing the Roles and Purposes of Various Professions” were the stationary success factor for adopting VM. This research establishes a road map for successful VM implementation via VM CSFs in Egypt and other underdeveloped nations. Stakeholders in the residential construction sector would benefit from this study by learning more about VM CSFs and how they may be used to increase the value of their projects

Introduction

Lines 73-78 introduce the topic of VM for maritime construction, maybe this data is not relevant for Egyptian residential construction. Moreover, it is not clear if the sentence beginning on line 75 is linked to the previous one on maritime construction or with the one before.

Many thanks for the reviewer’s valuable comment. We agree with the reviewer, the unwanted sentences have been deleted.

Please avoid exclamation marks ad the one between brackets on line 88.

Many thanks for the reviewer’s valuable comment. Exclamation marks have been mitigated

On line 89 virtual reality (VR) is mentioned for the first time, but it seems disconnected from what said until this point. A better introduction and explanation of the benefits of VR could help the reader to better follow the text logic.

We are sorry for this mistake. This should be VM instead VR. Consequently, we modified it.

Sentence on lines 100-101 talks about consensus on the definition of VM CSFs in residential construction projects. This sentence comes after the suggestion that CSFs fluctuate from one country to another (lines 99-100). Please better state how these conflicting statements are linked.

Many thanks, the meaning has been enhanced and corrected.

On line 104 you use the term "revolutionary". Since the research is site specific maybe it would be better to tone down this affirmation.

Many thanks, we agree with the reviewer, the meaning has been enhanced and corrected.

Lines 122-130 are not clear from a reader's point of view. Maybe a general rephrasing of the sentences would help in an easier reading. E.g., declare which is the latter factor mentioned on line 125 and state that "these phases" (line 129) are all the six previously mentioned and not only the three made explicit.

Many thanks, we agree with the reviewer, the meaning has been enhanced and corrected.

On line 166 data about UK are introduced. Since CSFs change depending on the Country, are these data necessary? In the same sentence, on line 167, "VM" is used to shorten "virtual machine" but it can create misunderstanding. Please consider of keeping "virtual machine management" without "(VM)".

Many thanks, we agree with the reviewer, the meaning has been enhanced and corrected.

Methodology

On line 192 you write "The Pearson correlation analysis was invited". Is "invited" the best term to use?

Many thanks, we agree with the reviewer, the meaning has been enhanced and corrected. The word has been replaced with employed

Please check consistency of CSFs Subscales names between Table 1 and Table 2. Maybe Table 1 should be better placed between section 3.1 and 3.2, since it refers to section 3.1 Semi-structured interviews.

Many thanks, we agree with the reviewer, Subscales names now are the same in both Tables. Also, Table 1 be placed between sections 3.1 and 3.2 based on the reviewer valuable comments.

Perhaps Figure 1 could supplement a portion of text explaining the method at the end of the part of section, rather than having the function of explaining the research method.

Many thanks for the reviewer’s comment. The figure now divided into steps and phases.

In section 3.1 the interviews process is explained. The ten interviews mentioned on line 203 were each undertaken by each of the fifteen experts? What about the 355 experts mentioned in the abstract?

Yes, the expression expert has been replaced with professionals to reflect the good meaning as the 335  are based on a questionnaire survey and 10 are based on the interview.

On line 216 you mention that three new factors were added. Which factors were added? Base on what were these new factors added?

These factors have been added based on the interview session.

Are references on line 222 correctly placed?

Yes

On line 231 you stated that 230 entities were created, but the proposal was only supported by 215: why?

Yes, as some entities did not complete the survey.

Check formatting of formula (1).

Corrected

The equation after line 276 is numbered (5) as the one before. Please check correct numbering.

Corrected

Data analysis

Check section title formatting on line 278.

Corrected

On line 282 "Table VII" is mentioned. Is it "Table 2"? If no, please check that "Table 2" is correctly mentioned in-text.

Corrected

On line 295, 34 VM CSFs are mentioned. Does it refer to Table 1? If yes, maybe mention it since the total number was not mentioned in section 3.1 Semi-structured interviews.

Mentioned

On line 306 there is a reference to "Table VI".

Corrected

Table 3, on line 333, should be Table 5. Check the formatting since part of the table is missing. Same for the following table.

Corrected

Discussion

On line 348 the sentence ends with colons followed by sub-sections. Maybe you can complete the sentence after colons to introduce the logic of the following sub-sections.

Corrected and sentence completed

Conclusions

On line 441 the analysis of 214 samples is mentioned, but this number of samples was not previously mentioned. Maybe can integrate it before in the appropriate section.

Corrected

Managerial implications

The presence of sections after the Conclusion is quite uncommon. Maybe this section could integrate the discussion section, and the new title for the section could be "5. Results and discussion".

Many thanks for the reviewer comment. We agree with the reviewer the above-mentioned section moved to the discussion section 

Theoretical implications

As per the previous section, it is quite odd to have other separate sections after the conclusions. Maybe this section can be part of the "6. Conclusion". Usually there is no presence of references at this point of papers, please avoid using them here.

Many thanks for the reviewer comment. We agree with the reviewer the above-mentioned section moved to the discussion section 

General comments

Some steps should be better explained in the methodology to allow the replication in other research.

Please check if the tables are correctly numbered and mentioned in-text.

References are fairly recent, almost 1/3 is from the past 5 years. Please check for consistency in the used style.

Thank you very much for reviewing our manuscript. We also greatly appreciate the reviewers for their complimentary comments and suggestions. The methodology now divided into steps however, all Tables and Figures are checked carefully. In addition, all references now are updated based on the reviewer valuable comment

Reviewer 3 Report

Dear Authors,

Please find my comments on the manuscript as follows:

§  Despite being interesting, the study does not have a good structure. The lack of clear methods and a discussion section makes it impossible to objectively understand the goals and outputs of the research. The manuscript has to be modified to a classical research structure.

§  The paper's goal and the importance of the findings are not made obvious in the presentation.

§  Conclusion and discussion sections are brief and often descriptive. The manuscript fails to adequately explore the important findings in the context of the relevant literature. Be sure to answer the "so what?" question at some point in the discussion/conclusion section by critically discussing your study's findings in light of the relevant literature.

§  The methodology, which details the research techniques the authors utilized and why needs to be developed. Sections about the methodology need to be clarified and updated.

§  There are frequent grammatical errors in the current manuscript.

§  We can see the enormous amount of work, but this article does not offer any novel solutions or research techniques.

Author Response

- Despite being interesting, the study does not have a good structure. The lack of clear methods and a discussion section makes it impossible to objectively understand the goals and outputs of the research. The manuscript has to be modified to a classical research structure.

Thank you very much for reviewing our manuscript. We have carried out the modifications that the reviewers suggested and revised the manuscript accordingly in the word file.  Please find below a point-by-point response to the reviewers’ concerns. We hope that you find our responses satisfactory and that the manuscript is now acceptable for publication.

-The paper's goal and the importance of the findings are not made obvious in the presentation.

We agree with the reviewer. The importance of the findings has been highlighted under new sections based on the reviewer’s valuable comments. Kindly refer to the theoretical and practical implications sections:

“5.2 The managerial contribution of this study
Egyptian and stakeholders elsewhere in developing countries could use the VM CSFs from this study as a 'framework' to carry out VM more efficiently to reduce the cost and improve their projects' quality. Consequently, Egypt needs to implement VM to acquire sustainable value since the investment is regularly connected to building development [92]. The proposed VM implementation ‘framework’ would help Egypt’s ambition to have a sustainable economy and become one of the top 30 nations in the world in future [93]. Furthermore, the ‘framework’ proposed by this study can help en-courage the adoption of VM in other developing countries where residential building projects are implemented using similar strategies [94]. This is essential to these coun-tries since they have many financial challenges adapting their building projects [95]. This study can also donate to the exiting body of knowledge by offering three new CSFs which were not highlighted before in the previous VM studies. These might require  further analysis of the construction project: “Motivate VM team members to produce VM output”, “Innovative method of generating ideas” and “Improved rates of inno-vation and assessment through the use of cutting-edge technology”. Finally, while this study has significantly contributed to both the implication and our body of knowledge, such limitations overture chances for future research. The data analysis depends on two hundred and twelve respondents. Another important influence may be accom-plished by larger sample size. This research portrayed the three types of respondents (owner, consultant, and contractor) as a single, unified entity. Future studies will pur-sue to develop the relationship using causal and predictive analysis among the various user groups in the building industry.

By rearranging CSFs, stakeholders like project owners and contractors may produce a "road map" for more efficient VM implementation in their projects. Moreover, this re-organization can serve as a standard for establishing a practical framework for effec-tively transitioning construction actors across VM phases. This will replace the old en-vironmental and sustainable performance standards that have been in place since the Arab Spring in 2011 [96]. Since the economy is often linked to the argument for sus-tainable growth [92], Egypt must implement VM if it ever hopes to have a genuinely sustainable economy.
The 'road map' will help Egypt achieve its goal of becoming one of the world's top 30 economies and a leading tourist destination [93]. The 'road map' created from this research may also be used to stimulate the use of VM in other developing countries concerning sustainable construction projects [94]. This is especially crucial in under-developed nations, often hampered by factors like the need to pay astronomical sums of money to address environmental concerns [95]. Consequently, VM could provide these nations with the chance to include eco-friendly practices in their building pro-jects' conceptualization stages [13, 97]. However, the following areas in which this re-search significantly contributes have each got important ramifications for the con-struction industry:
•    To determine whether or not the VM standards can remain competitive and suc-cessful in a global market, this database covers the VM standards and the related criteria.
•    It helps owners, consultants, and contractors evaluate and pick virtual reality (VR) deployment to improve construction projects' consistency, efficiency, and plan-ning.
•    It presents evidence from the scientific community that might help Egypt and oth-er developing nations to embrace VM.
•    Virtual reality (VR) and VR research in the field of construction have primarily focused on developed countries, and some few emerging countries (Malaysia, China, and Saudi Arabia, for example). Consequently, there is a lack data con-cerning VM adoption in developing countries and particularly analysis of VM's use in Egypt's construction sector. Therefore, our study has effectively established a link between VM and the Egyptian construction sector. Hence the study has laid a solid foundation on which to continue discussing the use of VM to increase the dependability of local construction projects and reduce the knowledge gap.
•    The results of this study can help advance the use of virtualization technology in construction in Egypt. This study has explained why organizations should adopt virtual machines. Benefits includes saving money and ensuring resources are allo-cated reasonably amongst projects. Thus, by designing and implementing the in-tended methods, all stakeholders may concentrate on the project's goal in terms of cost, timeliness, and efficacy. Achieving a high level of sustainability in a project has good consequences in the long run.
•    The findings of this study can also be used as a standard to measure the success of a project's execution and identify areas where improvements can be made. Issues including budget overruns, completing projects on time, and vague requirements all fell under this category. Additionally, business owners and managers could have access to information from this study that will help them implement VM in ways that increase the success of their enterprises.
•    Critical success factors (CSFs) for VM deployment were found, as were other VM-related criteria not included in earlier research.
5.3 Theoretical implications
Although research on sustainable concepts is not new [98], it is becoming increas-ingly important to many businesses [99]. The suggested prioritization approach stipu-lates the need for VM deployment, particularly around green home construction. Through the suggested methodology, this research discovered the CSFs for VM de-ployment. The challenges of deploying VM in Egypt's construction sector may be overcome with the help of these CSFs. Also, this research will help bridge the gap be-tween VM theory and practice. But as far as we are aware, no research has been done to look at the CSFs of VM deployment in the Egyptian construction industry. In the first place, this research provides an empirical identification of the crucial CSFs of VM that might facilitate the adoption of VM in the building sector. This discovery lays the groundwork for scholars, especially those in construction management, to investigate the CSFs of VM in underdeveloped nations.”

- Conclusion and discussion sections are brief and often descriptive. The manuscript fails to adequately explore the important findings in the context of the relevant literature. Be sure to answer the "so what?" question at some point in the discussion/conclusion section by critically discussing your study's findings in light of the relevant literature.

Many thanks for the reviewer’s valuable comment. The discussion and findings have been refined and highlighted the impact of the study based on the reviewer’s valuable comments. Kindly refer to the discussion section.

“5. Discussion
5.1 VM CSFs 
Pearson correlation analysis was used to validate the results and subscales ob-tained from the pilot study phase. Consequently, the VM CSFs executed were extracted under “Knowledge and Stakeholders”, “Environment and Culture”, “Dynamics of Workshop”, and “Standardisation”. This finding agreed with the finding regarding VM exploring CSFs in the exiting literature. For instance, according to Tanko, et al. [84], in the Nigerian construction industry, CSFs can be categorized under “People”, “Government”, “Environment”, and “Information/Methodology”. In this study, an analysis of the relative significance (RII) of the extracted groups and their components was performed. This is further discussed below:
5.1.1 Knowledge and Stakeholders
The first factor that contributed to the success of VM deployment was connected to “knowledge and Stakeholders”, the RII for “Ability to use and learn about VM” (RII = 0.82, SD = 1.09) and “Professional knowledge and expertise in the subjects of the par-ticipants” (RII = 0.80, SD = 1.17) have the maximum level. The bottom mean value be-longs to the items “Consumer Involvement” with (RII =0.71, SD=1.24) and “Willing-ness to accept changes and innovations” with (RII =0.72, SD=1.29). An RII of 0.738 in-dicates that all indicators are correspondingly significant. It further indicates that the level of stakeholder and knowledge-related success elements during VM deployment was more significant than the median of the scale (High-Medium level), which is bet-ter than the sufficient level. These results concurred with Tanko, et al. [29]. Their study argued  that the factors related to individuals and stakeholders would be crucial to efficiently enable the VM study with the required experience, as VM needs people and stakeholders. It is a proactive and innovative approach. These stakeholders are in-volved in the dynamic procedure that demands their obligations [85] and an active contribution to achieving the objectives of the workshop [86, 87]. Realizing these goals may ultimately aid in the project's development in both direct and indirect ways. Hence, Fong, et al. [88] posited that the ever-changing nature of projects in recent years necessitates a collaborative effort from all parties involved to create a project value team with creative solutions and insights.
5.1.2 Environment and Culture 
"Identifying and articulating the core values of a target audience" ranks highest on the "Culture and environment" dimension of the CSFs for VM implementation. It has a mean RII of 0.73 and a standard deviation of 1.06.  The lowest mean value belonged to “Encourage the VM team to provide VM deliverables” (RII =0.73, SD=1.10). Addition-ally, there was an average relevance of RII =0.732 across all indicators,. It shows that the level of this group during VM deployment was above the median of the scale (high-medium level), showing a more significant than the moderate level for this indi-cator. This finding is in line with the Tanko, et al. [29] who argued  that the "environ-ment" component encompasses the innovative ways in which VM members think and the ability to identify, define, and categorise the goals of construction projects through a collaborative and problem-solving approach. The “Culture and the Environment” comprise the situations and environment in which stakeholders operate to enable ef-fective interactions and working relations [39]. Collaboration and cooperation among VM team members will promote innovation and creative alternatives for problems [17, 60, 61]. Consequently, the organisation's value system will improve, and the client will be more convinced to implement VM [62].
5.1.3 Dynamics of workshop
Dynamics of workshop " was the subject of the third subscale of the CSFs for the VM implementation.” the RII value for “Gathering of Contextual Data” has the high-est value with (RII =0.73, SD=1.03). The lowest mean value belongs to two items, “Awareness on the part of clients concerning value optimization role of VM” with (RII =0.7, SD=1.00) and “VM feedback mechanism” with (RII =0.68, SD=1.10). Across all parameters, RII = 0.722 which has indicated a high level of significance. It suggests that stakeholder interest and expert knowledge were significantly greater than average during the VM implementation process (high-medium level). Ramly, et al. [25] re-vealed  results from several studies indicating that the structured procedures and the work plan constitute VM's core values that distinguish them from other management techniques. Researchers believe each professional must prepare information on how the project relates to them in the VM study [29]. The methodology is a procedure that should be performed to execute a project [29]. Hence, Ramly, et al. [25] argued that the VM study's best practice is the commitment of the team leader to facilitate the proce-dures under the VM work plan. However, the design team can devise numerous ways to enhance the project through workshop [59]. In the same vein, Coetzee [89] posited that technical advancements should be considered and utilized in VM activities. Using electronic devices to establish VM will save money and time by eliminating the need to bring together physical team members [90]. Moreover, it will enable a virtual VM en-vironment by rewarding professionals with the requisite competencies and skills [91].
5.1.4 Standardization
The " VM study plan for execution" RII (RII = 0.75, SD = 1.09) was the highest score for the "Standardization" subscale of the CSFs of VM implementation. The median value of "The client's tenacity in conveying their demands and constraints to the de-sign team" is 0.69 (RII = 0.69, SD = 1.00). All indicators were given an aggregate signif-icance of RII = 0.7. This indicator is more significant than average because the stake-holder and expertise factor levels for successful VM implementation are above the me-dian value. It is the major consumer and investor, with much of the capital formation linked to investment in properties and infrastructure, the government should produce all VM's policies [84]. Therefore, it would be crucial to have the government's backing and involvement in imposing the VM on modern building practices [84].”

- The methodology, which details the research techniques the authors utilized and why needs to be developed. Sections about the methodology need to be clarified and updated.

Thank you for your valuable comment. The methodology section has been illustrated and divided into steps to highlight the research phases and clarified well based on the reviewer’s valuable comments. Kindly refer to the methodology section:

Methodology
The aim of this study is to recognise the critical VM CSFs in Egyptian construction. Thus, the research methodology summarised the procedures adopted to achieve this aim [52]. This was accomplished through reading relevant material and conducting a semi-structured interview. A pilot study through Exploratory Factor Analysis (EFA) was performed to check the results obtained from the interview [53]. Therefore, the questionnaire survey was conducted to ask professionals with sufficient VM studies experience to express their opinion on each of the nominated factors. The Pearson correlation analysis was employed to calculate the correlation among the factors and validate the results from EFA. Moreover, the Relative Importance Index (RII) analysis was conducted to examine the different factors and groups that are vital to implementing VM effectively in the Egyptian building industry. The research design, which is adapted from [54, 55] is illustrated in Figure1.

    Phase I          Phase II    Phase III
Figure 1: Research design
3.1 Semi-structured interviews
Based on the suggestions made by Sanders [56] and Hesse-Biber [57], the research involved ten interviews. Hence fifteen experts on three levels were selected: (i) the years of experience, (ii) the educational attainment, and (iii) position through a “purposive sampling” approach. Four professionals, five practitioners from the private sector and six professionals from the consultative industry were interviewed. The interviewees have vast experience in the residential building field ranging from nine and 40 years, and the contributors were selected based on the following criteria: experience, education, and work. Likewise four academics, five private industry practitioners, and six consultants were interviewed, whoheld several positions, including site engineer, consultant, project manager, executive director, and the manager following  Othman, et al. [58]. Their primary roles cover all key stakeholders, clients, suppliers, or contractors in the building industry, ensuring extensive experience from various perspectives. Their experience also included working with government, private sector and self-employed agencies. Consequently, the interviewed experts appealed that a more proper system must champion the VM implementation in projects and categorised VM CSFs into four categories, as shown in Table I [53]. In addition, three new factors were added to the list and several VM CSFs were modified, as shown in Table 1. [53]. The revised and new tasks were utilised to create the pilot study questionnaire.

Table 1: VM CSFs
CSFs Subscales (Groups)    Code    Name    Researches
Knowledge and Stakeholders     SF.SK1    Constructing a VM team from a variety of discipline    [15, 17, 23]
    SF.SK2    Competence of VM facilitator    [15, 23, 37, 59, 60]
    SF.SK3    Collaborative discussion that is well-communicated    [61, 62]
    SF.SK4    Capability to lead a VM workshop    [27]
    SF.SK5    Ability to use and learn about VM    [15, 17, 23]
    SF.SK6    Participation of all relevant parties in the VM workshop    [17, 61]
    SF.SK7    Professional knowledge and expertise in the subjects of the participants    [17]
    SF.SK8    Readiness to embrace novel ideas and approaches    [17]
    SF.SK9    Establishing the roles and purposes of various professions    [61]
    SF.SK10    consumer involvement    [23]
    SF.SK11    Competence and character traits of the individuals involved    [17]
    SF.SK12    Stakeholder and agency cooperation and a high-quality working relationship    [17, 59, 62]
    SF.SK13    Participant discipline and attitude    [23]
environment and Culture     SF.CE1    Workshop attendees articulated their VM's clear and defined purpose    [23, 62]
    SF.CE2    Participant organisations' delegation of decision-making authority    [23]
    SF.CE3    Identifying and articulating the core values of a target audience    [63]
    SF.CE4    Motivate VM designer to generate VM outputs    [64]
Dynamics of the Workshop     SF.WD1    A proactive, imaginative, and organised strategy    [15, 17]
    SF.WD2    Function and component analysis of the project    [17]
    SF.WD3    VM feedback mechanism    [27]
    SF.WD4    Customers' understanding of VM's value-optimization potential    [27]
    SF.WD5    Appropriate input from the original design team    [60]
    SF.WD6    VM workshop was appropriately timed.    [17]
    SF.WD7    Gathering of contextual data    [23]
    SF.WD8    Group orientation    [62, 65]
    SF.WD9    Innovative method of generating ideas    [64]
    SF.WD10    Improved rates of innovation and assessment through the use of cutting-edge technology    [64]
    SF.WD11    Integration of virtual reality workshops into the project lifecycle    [23]
Standardisation    SF.ST1    Clients' involvement and encouragement    [15, 17, 23, 62]
    SF.ST2    Suggestions from the proper state and municipal agencies    [37]
    SF.ST3    Consistent presence of the policy maker    [15]
    SF.ST4    VM workshop strategy for execution    [15, 17, 37]
    SF.ST5    An official government promise to adopt VM    [6]

3.2 Pilot survey
A pilot study in the Egyptian residential construction industry was undertaken to explore the results mentioned above through EFA, which sent the pilot questionnaire to an appropriate number of participants (200 construction professionals) [53, 66].  The research instrument’s reliability was tested usingthe Cronbach alpha test. This test enables assessing the reliability of the in-area questionnaire and all the fields considered. The alpha values obtained ranged from 0.84 to 0.91, indicating a high-reliability level for the study’s surveys [53, 67].
3.3 Main survey
As VM implementation is relatively new in Egypt, a stratified sampling of the specific subspecies has been considered [23]. Stratification considers demographic variations in the three industries (client, consultant, and contractor) [68]. The screening study created over 280 entities, although the proposal was only supported by 215. The survey was used to assess the level of VM implementation, awareness and identify essential CSFs using the research instrument (Questionnaire) recommended by Fellows and Liu [69]. Consequently, the results show that the participants have enough VM awareness and knowledge.
3.4 Pearson Correlation Analysis
The aim of this study is to identify the critical VM activities in the Egyptian construction industry. Consequently, it is essential to check the correlations between new data from the primary survey. In the natural sciences, the Pearson correlation factor is widely used [70]. It is used to calculate the correlation among two variables X and Y, whose estimates are between -1 and 1 and calculated by the following equation:

                                                                                                                  (1)

where X¯ = mean value of Sample one; Y¯ = mean value of sample two; and r represents the Pearson correlation coefficient. The estimated value range of r is from -1 to 1. The greater the absolute value, the greater the degree of correlation. The higher the coefficient of correlation is to 1 or -1, the greater the degree of correlation. conversely, the quieter the coefficient of correlation to 0, and the lower the correlation. This was used to explain the association between VM CSFs groups of the Pearson coefficient. The correlation of these groups was determined automatically using the Statistical Package for the Social Sciences (SPSS) software.
3.5 Ranking analysis
Relative Importance Index (RII) is the most commonly used method for rankings of the attributes [71, 72] as identified by Salleh [73] as statistical method used to identify ranks of different causes. The response events' frequency and intensity were evaluated in Equation 1 [74, 75], using 5-point Likert scale and RII.
RII=(∑â–’w)/(A×N)=(5n_5+4n_4+3n_3+2n_2+1n_1)/(5×N)    (2)

where W indicates the respondent's weighting to each variable, A is the maximum weight, and N is the whole number of members. Table 4 shows the results of RII ranks. This calculation was furtherbe classified using the three selected respondents’ groups (owner, consultant, and contractor) to cross-compare the relative significance of the factors perceived by the selected three groups. Using this assessment, the study can identify the most critical VM CSFs contributing to VM implementation in Egypt’s residential building industry.
3.6 Stationary analysis (Ginni’s Mean)
To determine the VM CSFs, our study followed Samuel and Ovie [76]’s strategy. The following are the steps involved in this strategy:
    As stated in equation (3), “Ginni's mean difference measure of dispersion” [77] may be used to calculate the average spread of the RII values.
G.M=G/M    (3)
Ginni's mean difference (G.M) is a measure of dispersion where N is the number of factors and G is the sum of the changes in value between all imaginable pairs of variables, and M is the total number of variances
M=(N(N-1))/2    (4)
b) Equation (5) is used to calculate weights for each RII number based on the predicted Ginni's mean difference measure of dispersion:
Wi=G.M×RIIi/RII1    (5)
where RIIi is the relative index number of any CSFs, RII1 is the greatest relative index number, and Wi is the weight of each RII number. 
c) RII central value can be represented by approving the geometric mean (G.M. (w)) of the RII numbers and by fitting this value to the RII calibration to reflect the stationary, as defined by  equation (6):
G:M.(w)=Antilog (∑▒〖w.logRII〗)/(∑â–’w)
(6)
where ∑w: is equal to the weights given to the RII numbers as a whole.

- There are frequent grammatical errors in the current manuscript.

We agree with the reviewer. The whole language of the paper is enhanced according to the reviewer’s opinion.

- We can see the enormous amount of work, but this article does not offer any novel solutions or research techniques.

Many thanks for the reviewer’s valuable comment. We also greatly appreciate the reviewer for their complimentary comments and suggestions. The current form has been enhanced more to reflect the novelty of the paper.

Round 2

Reviewer 2 Report

The revisions made by the authors to the comments received in the previous round of reviews have improved the clarity of the paper.

Some of the suggestions were not considered, even if differently stated. In future, please consider of explaining why you prefer not to make changes on your paper rather than responding that changes were made based on reviewers' suggestions.

Please make sure that your tables numbering and formatting is correct since there still are some errors.

I would suggest a final throughout review of the paper before the publishing.

Author Response

The revisions made by the authors to the comments received in the previous round of reviews have improved the clarity of the paper. Some of the suggestions were not considered, even if differently stated. In future, please consider of explaining why you prefer not to make changes on your paper rather than responding that changes were made based on reviewers' suggestions.

Please make sure that your tables numbering and formatting is correct since there still are some errors.

I would suggest a final throughout review of the paper before the publishing.

Thank you very much for reviewing our manuscript. We have carried out the modifications that the reviewers suggested and revised the manuscript accordingly in the word file. We hope that you find our responses satisfactory and that the manuscript is now acceptable for publication. In addition, the whole paper proofread and the above-mentioned errors have been corrected. Also, the whole Tables and Figures have been also checked.

Again, Many thanks for the reviewer valuable comments and recommendations  

Reviewer 3 Report

Dear Authors,

Thank you for the amendments to the manuscript. There are a few more comments listed below:

1.     Minor grammatical mistakes are still in the manuscript. For instance, (L18)  “A total of 335 professionalsin the residential con- 18 struction industry in Cairo and Giza filled out the questionnaire, and 200 had completed the pilot 19 questionnaire”. It is important to check the manuscript for grammatical errors thoroughly.

2.     The presentation of "Figure 1: Research Design" can be improved.

Author Response

Thank you for the amendments to the manuscript. There are a few more comments listed below:

  1. Minor grammatical mistakes are still in the manuscript. For instance, (L18)  “A total of 335 professionalsin the residential con- 18 struction industry in Cairo and Giza filled out the questionnaire, and 200 had completed the pilot 19 questionnaire”. It is important to check the manuscript for grammatical errors thoroughly.

Thank you very much for reviewing our manuscript. We have carried out the modifications that the reviewers suggested and revised the manuscript accordingly in the word file. We hope that you find our responses satisfactory and that the manuscript is now acceptable for publication. In addition, the whole paper proofread and the above-mentioned statement has been enhanced and corrected kindly refer to the abstract section:

“During the past two decades, value management (VM), has developed into a recognized construction practice. However, the methods and activities associated with VM adopt informal approaches in developing countries. This study aims to explore the Critical Success Factors (CSFs) of VM implementation. Consequently, VM CSFs were investigated from previous literature and fur-ther categorized over a semi-structured interview. The importance of these CSFs investigated by 335 structured questionnaires by residential building professionals. Subsequently, the exploratory study using the Exploratory Pearson correlation of the VM CSFs was employed to validate the categorization resulting from a semi-structured interview and pilot study phases. Based on the validation results, the VM CSFs may be divided into four dimensions: culture and environment, workshop dynamics, stakeholder and knowledge, and standardization. Through Important Relative Index (RII) analysis, the essential CSFs are Creating a VM Team from a Variety of Discipline, VM knowledge, experience of participants, and professional experience of the different participants' diverse disciplines. In addition, this research used a stationary analytic strategy to evaluate the degree to which VM Critical Success Factors (CSFs) have been incorporated into residential construction projects in Egypt. The results revealed that “Establishing the Roles and Purposes of Various Professions” were the stationary success factor for adopting VM. This research establishes a road map for successful VM implementation via VM CSFs in Egypt and other underdeveloped nations. Stakeholders in the residential construction sector would benefit from this study by learning more about VM CSFs and how they may be used to increase the value of their projects.

  1. The presentation of "Figure 1: Research Design" can be improved.

Many thanks for the reviewer valuable comment. The better presentation for the research design has been added.
